# Oxidation of *Microcystis aeruginosa* and Microcystins with Peracetic Acid

**DOI:** 10.3390/toxins16080328

**Published:** 2024-07-23

**Authors:** Mennatallah Alnahas, Husein Almuhtaram, Ron Hofmann

**Affiliations:** Department of Civil and Mineral Engineering, University of Toronto, Toronto, ON M5S 1A4, Canada; husein.almuhtaram@utoronto.ca (H.A.); ron.hofmann@utoronto.ca (R.H.)

**Keywords:** preoxidation, cyanobacteria, cyanotoxins, harmful algal blooms, drinking water treatment

## Abstract

Peracetic acid (PAA) shows potential for use in drinking water treatment as an alternative to prechlorination, such as for mussel control and disinfection by-product precursor destruction, though its impact as a preoxidant during cyanobacterial blooms remains underexplored. Here, *Microcystis aeruginosa* inactivation and microcystin-LR and -RR release and degradation using PAA were explored. The toxin degradation rates were found to be higher in alkaline conditions than in neutral and acidic conditions. However, all rates were significantly smaller than comparable rates when using free chlorine. The inactivation of *M. aeruginosa* cells using PAA was faster at acidic pH, showing immediate cell damage and subsequent cell death after 15–60 min of exposure to 10 mg/L PAA. In neutral and alkaline conditions, cell death occurred after a longer lag phase (3–6 h). During cell inactivation, microcystin-LR was released slowly, with <35% of the initial intracellular toxins measured in solution after 12 h of exposure to 10 mg/L PAA. Overall, PAA appears impractically slow for *M. aeruginosa* cell inactivation or microcystin-LR and -RR destruction in drinking water treatment, but this slow reactivity may also allow it to continue to be applied as a preoxidant for other purposes during cyanobacterial blooms without the risk of toxin release.

## 1. Introduction

Drinking water utilities that draw supplies from surface water resources often apply oxidants such as chlorine at the point of intake to protect against invasive mussels, suppress biofouling within the plant, or lyse cyanobacteria and degrade cyanotoxins. The use of chlorine, however, may result in the formation of regulated disinfection byproducts (DBPs), so it may be desirable to use non-chlorine oxidants if they can achieve the same objectives. Peracetic acid (PAA) is a promising alternative that has been reported to be effective at suppressing mussel growth [1] and destroying DBP precursors [2], and it therefore might warrant more widespread consideration as a preoxidant. However, it is unknown if it might lyse cyanobacteria cells if they are present in the source water, potentially releasing cyanotoxins into the water, or conversely, if it might be able to effectively destroy cyanotoxins if present.

Commercial PAA-based products range in concentration from 5% to 40% PAA in equilibrium solution with acetic acid and hydrogen peroxide (Equation (1) [3]). PAA has a relatively high oxidation-reduction potential (1.38 V), and given a pKa of 8.2, it exists as both peracetate and peracetic acid at the typical pH of drinking water [4,5]. Hydrogen peroxide (H_2_O_2_) alone is considered a weak oxidant, but it has been demonstrated to lyse cyanobacteria cells when applied for a prolonged period (days) [6,7,8,9,10].
(1)$$CH_{3}-COOH+H_{2}O_{2}\overset{H^{+}}{↔}CH_{3}-COOOH+H_{2}O$$

In the context of cyanobacteria and cyanotoxin control, studies investigating PAA are scarce. One study investigated the use of PAA as an algaecide for aquacultures experiencing cyanobacteria blooms. After 1 day of exposure to 10 mg/L of PAA-based products, they observed a significant reduction in phycocyanin, the characteristic cyanobacterial photosynthetic pigment, in water samples extracted from aquaculture ponds dominated by *Microcystis* [11]. Another study reported that exposure to 10 mg/L of PAA for 20 min caused insignificant damage to cell membranes of *Microcystis aeruginosa* (Kützing, 1846) with a cell concentration of 2.2 × 10^6^ cells/mL [12]. Similarly, Cao et al. [13] reported a reaction rate of 0.03 × 10^−9^ min^−1^ at a neutral pH for the inactivation of 1.3 × 10^6^ *M. aeruginosa* cells/mL using 7.6 mg/L PAA, which translates to <2% cell inactivation in 20 min. For *M. aeruginosa* in the order of 10^6^ cells/mL, resistance to oxidation by PAA may be due to an extracellular polysaccharide mucilage that surrounds cell colonies [14]. In contrast, at pH 4.3 and for a lower cell concentration of 1 × 10^5^ *M. aeruginosa* cells/mL, the release and subsequent 1-log reduction in microcystin-LR (MC-LR) has been reported within 45 min following oxidation by 10 mg/L PAA [14]; however, cell lysis was not directly measured in that study but rather estimated based on the release of intracellular MC-LR, which does not consider the potential lysis of non-toxic cells or account for the cumulative damage that cells can sustain before lysing over a longer period of time [6], [13]. The potential for toxin release following oxidation was not investigated by Zhu & Liu [12] or Cao et al. [13]. Overall, cell lysis rates and the potential for toxin release following PAA oxidation in conditions typical of drinking water treatment are not well established in the literature.

There is also little information about the direct rate of reaction between PAA and cyanotoxins. At pH 4.3, the reaction rate coefficient between PAA and MC-LR has been reported to be 3.46 M^−1^ s^−1^, which is two orders of magnitude lower than that of hypochlorous acid at the same pH (475 M^−1^ s^−1^) [14,15]. The reaction rate has not been investigated at pH typical of drinking water, limiting the ability of stakeholders to make well-informed decisions about the application of peracetic acid during active cyanobacterial blooms.

The objectives of this study are to (1) investigate the impact of PAA on cyanobacteria cells in terms of cell damage and death, as well as the release and degradation of cell-bound cyanotoxins under conditions typical of drinking water, and (2) investigate the ability of PAA to degrade MC-LR and -RR in the absence of cyanobacteria cells. The significance of this study lies in addressing the unclear impact of using PAA as a preoxidant in drinking water treatment plants encountering cyanobacteria blooms, as pre-existing research did not examine cyanotoxin release and degradation under typical treatment conditions including pH levels and contact times.

## 2. Results and Discussion

### 2.1. Degradation of Dissolved MC-LR and MC-RR by PAA

The degradation of 50 µg/L of dissolved MC-LR/RR by 10 mg/L PAA solution (including H_2_O_2_) at pHs 6 and 7 was insignificant (<10%) over 4 h (Appendix A). At pH 8, 30% and 20% reductions in MC-LR and MC-RR were observed, respectively, suggesting that PAA is more effective at degrading microcystins at higher pHs. Greater PAA reactivity at higher pHs has also been reported in terms of the destruction of methionine [16], as well as trihalomethane precursors [2], and may reflect a greater reactivity of the peracetate ion form of PAA, which becomes more prevalent at pH values approaching and above the pKa. The second-order reaction rate coefficients between PAA and MC-LR and -RR (Table 1) are 2 to 5 orders of magnitude lower than the reported values for chlorine under the same conditions [15]. In order to isolate the effect of the co-existing H_2_O_2_ in the PAA solution on toxin degradation, we also investigated the rate of degradation of 50 µg/L of dissolved MC-LR/RR using H_2_O_2_ alone at pH 8. Our results indicate that the role of H_2_O_2_ is negligible in MC-LR degradation (Appendix A), which is consistent with earlier findings by Ren et al. [17]. For MC-RR, H_2_O_2_ contributed approximately 50% of the degradation observed with the PAA solution. However, microcystin degradation rates by both oxidants are negligible when compared to free chlorine. These findings indicate that PAA likely does not destroy dissolved microcystins when considering dosages and contact times relevant for preoxidation in actual practice.

### 2.2. Inactivation of M. aeruginosa by PAA

#### 2.2.1. Acidic Conditions

The inactivation of $1\times 10^{6}$ cells/mL *M. aeruginosa* by 10 mg/L PAA solution under acidic conditions is shown in Figure 1a,b. At pH 4.4, there is a delay of approximately 15 min between the addition of PAA and the death of *M. aeruginosa* cells, at which point the PI stain becomes dominant (Figure 2a). During this lag phase (e.g., at 7.5 min), signs of partial cell damage are indicated by the detection of both FDA and PI stains (Q2) in >25% of cells (Figure 3). The lag-phase phenomenon has been reported when using other oxidants including chlorine, hydrogen peroxide, and permanganate and is attributed to cell membranes being able to withstand a certain level of oxidation prior to disintegrating [18,19,20]. The lag phase is followed by an increase in the percentage of dead cells and a corresponding decline in both live and damaged cells over the following 45 min, reaching a 1-log reduction in cells by the end of the trial (Figure 1a). In contrast, PAA was less effective at degrading cell membranes at pH 5.2 because the lag phase persisted beyond the first hour of contact time (Figure 1b). Following 2 h of exposure, the percentage of dead cells plateaued at approximately 90%, and the remaining cells were in a damaged state (Figure 1b). The decay in PAA was insignificant throughout the period of the trials under these acidic conditions (Figure 1a,b). While these results demonstrate that PAA can lyse *M. aeruginosa* cells under acidic conditions, such a low pH is likely not directly relevant to preoxidation in a drinking water treatment context.

#### 2.2.2. Neutral and Alkaline Conditions

Under neutral and alkaline conditions (pH 7.0 and 8.5), cell inactivation was slower than under acidic conditions but followed the same pattern whereby cell wall damage accumulated prior to the onset of cell death (Figure 1c,d). At pH 7.0, cell death was not observed in the first three hours of exposure; however, approximately 50% of total cells had signs of damage (Figure 2c). The percentage of dead cells did not increase beyond 65% over a 6 h reaction time (Figure 2c). At pH 8.5, *M. aeruginosa* cells remained susceptible to cell wall damage but only 47% were dead by the end of the 24 h exposure time. The preoxidation of toxic cyanobacteria using PAA may cause damage to cell membranes, potentially leading to microcystin release without subsequent degradation. Thus, the application of PAA as a preoxidant in drinking water treatment may not be advisable during an active bloom, unless it can also destroy released toxins (which it likely cannot, as discussed later).

The rate of PAA decay under neutral and alkaline conditions was higher compared to the observed decay under acidic conditions (Figure 1c,d). At the end of the 24 h trial period, PAA had decayed by 85% and 97% at pHs 7 and 8.5, respectively, compared to negligible decay at pHs 4.4 and 5.2. Spontaneous decomposition of PAA is negligible under acidic conditions (pH < 6) [21], whereas the decomposition of PAA under neutral and alkaline conditions is catalyzed by several factors including OH^−^ and organic matter [21,22,23]. The observed decay of PAA in neutral and alkaline conditions was faster compared to the spontaneous decomposition kinetics in ultrapure water reported by Chen et al. [24] (Appendix A). This can be attributed to the oxidative demand by *M. aeruginosa* and the released intracellular organic matter.

### 2.3. Kinetics of the Reaction of PAA with M. aeruginosa

Reaction kinetic models are necessary to calculate the exact doses, in terms of concentration and contact time, for PAA to accomplish a desired level of cell or toxin control in drinking water treatment. Furthermore, fitting observed data into reaction kinetic models also provides a base for comparing PAA efficacy to other oxidants. Because a lag phase was observed prior to the onset of cell death, the delayed Chick–Watson model (Equation (2)) was used to fit the PAA inactivation kinetics in terms of cell viability. Cell viability is defined in terms of the detection of enzymatic activity (FDA-stained), where $\frac{N_{V_{CT}}}{N_{V_{0}}}$ is the ratio of viable cells ($N_{V_{CT}}$) at time *T* to the initial concentration of viable cells ($N_{V_{0}}$), $T_{lag}$ is the lag phase time, *C* is the concentration of PAA, and $k_{1}$ is an empirical reaction rate coefficient.
(2)$$\ln\frac{N_{V_{CT}}}{N_{V_{0}}}=\left\{\begin{array}{l} \;\;\;0\;\;\;\;\;\;\;\;for\;CT\leq {CT}_{lag}\\ k_{1}(CT-{CT}_{lag})\;\;for\;CT>{CT}_{lag} \end{array}\right.$$

#### 2.3.1. Effect of PAA Dosage

To understand the rate of inactivation of *M. aeruginosa* by PAA, different PAA dosages were applied at pH 5.2. The impact of PAA dosage was not investigated under alkaline and neutral conditions due to the observed slow inactivation rates at a high PAA dosage (10 mg/L). According to an analysis of covariance (ANCOVA), the effect of PAA dosage is less significant than that of the interaction of concentration with time (CT) (*p*-value < 0.001), implying that the increase in the rate of cell degradation observed with increasing PAA dosage can be attributed mostly to the accumulated CT. The rate of inactivation of $1\times 10^{6}$ cells/mL *M. aeruginosa* with PAA at pH 5.2 was therefore fitted to the delayed Chick–Watson model using CT rather than PAA concentration only. The model was used to determine ${CT}_{lag}$ at pH 5.2, represented by the red dotted line using different initial PAA dosages (Figure 3). The observed reaction rate coefficient $k_{1}$ is $8.8\times 10^{-4}\;{\left(\frac{mg}{L}\right)}^{-1}{\cdot min}^{-1}\;$(R^2^ = 0.88), which expresses the rate of cell death following a ${CT}_{lag}$ of $240\frac{mg}{L}\cdot min$. The ${CT}_{lag}$ represents the PAA exposure required to accumulate damage before cell death is observed. The inactivation rates reported in this study do not isolate the role of H_2_O_2_ because its previously reported reaction rates are negligible, as shown in [6,13]. At pH 7, Chang et al. [25] reported that a $T_{lag}$ of 109 min occurs prior to *M. aeruginosa* cell rupture when subjected to 2.5 mg/L of H_2_O_2_, whereas the current study shows that >30% of live cells were damaged within 30 min using the PAA mixture (10 mg/L PAA and 1.2 mg/L H_2_O_2_). This implies that the primary factor contributing to the inactivation of *M. aeruginosa* in the PAA solution is PAA rather than the co-existing H_2_O_2_.

#### 2.3.2. Effect of pH

PAA reaction with *M. aeruginosa* cells was observed to be slower under neutral and alkaline conditions compared to acidic conditions (Figure 4), with the delayed Chick–Watson model kinetics being reported in Table 2. The reaction coefficients are observed to decrease with increasing pH, while the *CT_lag_* needed before cell death begins to increase with pH. This is in agreement with the general trend of bacterial inactivation using PAA, where greater activity is observed at lower pHs [26]. For instance, higher inactivation with a lower pH has been similarly reported with *Escherichia coli* [27]. This might be related to the dissociation of PAA at pH 8.2, i.e., the pKa of PAA, since the dissociated form (CH_3_CO_3_^−^) is less reactive than its non-dissociated form (CH_3_CO_3_H) [26]. The observed trend of increasing *M. aeruginosa* inactivation with decreasing pH using PAA is similar to that observed with chlorine [28,29,30]. The higher chlorine reactivity at a lower pH is attributed to the dominance of hypochlorous acid (HOCl). However, the kinetics observed for the PAA inactivation of *M. aeruginosa* cells at pH 8.5 in this study is in the order of 10^4^ times slower than when using chlorine at pH 8.3–8.6, reported by Lin et al. [29].

The kinetics of PAA inactivation of cyanobacteria at different pHs have not been reported previously. However, at pH 7, Cao et al. [13] reported a 22% increase in *M. aeruginosa* cell permeability after 45 min following a 40 min lag phase using 100 µM PAA (7.6 mg/L) plus 84 µM H_2_O_2_ (2.8 mg/L). In contrast, a 50% increase in damaged cells (defined as cells with permeable membranes and enzymatic activity) was observed in this study after 60 min following exposure to 131 µM PAA (10 mg/L) plus 35 µM H_2_O_2_ (1.2 mg/L) with no lag phase. These results are relatively similar: requiring about an hour or two of exposure to about 100 μM PAA (7.6 mg/L) to damage about 20–50% of the cells at a neutral pH.

### 2.4. Microcystin-LR Release and Degradation by PAA

#### 2.4.1. Total MC-LR

The degradation of total MC-LR (extracellular and cell-bound) at pH 5.2 using different PAA dosages was explored (Figure 5a). It was observed that approximately half of the cell-bound and extracellular MC-LR was destroyed within about 6 h, with no statistically significant difference observed for the different PAA dosages ranging from 2 to 10 mg/L. The ability of PAA to degrade MC-LR under acidic conditions agrees with the 1-log destruction of released microcystin-LR observed at pH 4.2 in the presence of $1\times 10^{5}$ cells/mL *M. aeruginosa* by Almuhtaram and Hofmann [14]. Under neutral and alkaline conditions, 10 mg/L of PAA caused no statistically significant decrease in total MC-LR after even 12 h (Figure 5b), although there might have been a true (but slight) actual decay whose significance could not be established due to the variability in the data (see the large error bars). Nevertheless, any reactions at neutral and alkaline pHs are likely slow enough to be impractical as a means to control MC-LR by using PAA at 10 mg/L.

#### 2.4.2. Extracellular MC-LR

The release and degradation of initially cell-bound MC-LR from 1 × 10^6^ cells/mL *M. aeruginosa* was investigated. Prior to PAA exposure, the initial extracellular MC-LR concentration was approximately 3 ± 0.7 μg/L and the total (extracellular and cell-bound) concentration ranged between 85 and 140 μg/L, implying that almost all initial MC-LR was cell-bound.

Under acidic conditions (pH 5.2) and in the absence of PAA (i.e., the control), approximately 5 µg/L of MC-LR was released after 6 h (Figure 6a). For a PAA dosage of 2 mg/L, MC-LR release was not statistically different from the control, but a dosage of 5 mg/L led to an increase of about 15 µg/L in extracellular MC-LR after 6 h. A PAA dosage of 10 mg/L led to a 35 µg/L increase in extracellular MC-LR after 6 h. Note that during these 6 h, it can be assumed that around 50% of the released MC-LR might have been destroyed by the PAA (as shown in Figure 5a), so the actual release of MC-LR from the cells into the water might have been in the order of 30–70 µg/L for PAA dosages of 5 and 10 mg/L, respectively, over 6 h. This represents perhaps up to about half of the initial cell-bound MC-LR being released, presumably due to cell lysis by PAA.

MC-LR release under neutral and alkaline conditions was slower than at pH 5.2 (Figure 6b), consistent with the slower cell damage rates caused by PAA reported in this study. Approximately 35 µg/L of MC-LR was released under neutral and alkaline conditions in 12 h, which is double the time required to release the same amount of MC-LR under acidic conditions. At pH 7.0, 2-log reduction in live cells was observed following 6 h of contact time with PAA, but the percentage of MC-LR released relative to the total MC-LR concentration (extracellular and cell-bound) was <15% (<15 µg/L as extracellular MC-LR). Thus, it is evident that the damage to the cells caused by PAA did not trigger a rapid release of the cell-bound MC-LR at this neutral pH. The slow release of cell-bound MC-LR is mainly attributed to the mechanism of inactivation of *M. aeruginosa* by PAA at a neutral pH, where cells sustain damage to cell wall for hours before dying or lysing. By the end of 12 h of contact time, only 30% and 35% of the cell-bound MC-LR was released under neutral and alkaline conditions, respectively. Since the concentration of total MC-LR did not decrease significantly under these conditions (Figure 5b), the concentration of extracellular MC-LR measured can be considered to account for the total amount of MC-LR released from the cells.

## 3. Conclusions

This study emphasises that PAA may not be an effective oxidant for controlling either cyanobacteria cells or MC-LR and -RR in drinking water utilities, as the degradation rates of both the toxins and the cells by PAA were orders of magnitude smaller than free chlorine, especially at the neutral and alkaline pHs typical of drinking water. This does mean, however, that the use of PAA as an oxidant for other purposes, such as mussel control at the point of drinking water plant intake, may not pose a significant risk of lysing cells and releasing toxins under contact times and dosages typical of drinking water treatment, since such cell lysis would take hours at a high PAA dosage (e.g., 10 mg/L) at pH 7.0 and 8.5.

This conclusion reflects the presence of approximately 10% H_2_O_2_ in a PAA solution. H_2_O_2_ contributes only minimally to cell destruction and toxin degradation under the conditions studied here and that would exist in practice.

## 4. Materials and Methods

### 4.1. Cell Culture

A toxic strain of *M. aeruginosa* (CPCC 299) was obtained from the Canadian Phycological Culture Centre (Waterloo, ON, Canada), cultivated in batch cultures using sterile BG-11 medium and incubated under specific growth conditions (25 ± 1 °C with a 12 h diurnal cycle using white LEDs with an intensity of 13 μE/m^2^). *M. aeruginosa* cell suspensions were harvested in the exponential growth phase, and cell pellets were prepared in 45 mL aliquots. Aliquots were centrifuged at 1900 rpm for 14 min at 8 °C, the supernatant was decanted, and cells were resuspended with Tris-buffered Milli-Q water, three times. Cell concentrations were determined by counting in a Sedgewick–Rafter counting chamber using a Nikon Eclipse microscope at 10× objective magnification. Solutions were diluted in phosphate-based buffered Milli-Q water at pH 7.6 to reach a final cell concentration of 1 × 10^6^ cell/mL.

### 4.2. Analytical Methods

The concentrations of PAA and H_2_O_2_ were determined using the Hach N,N-diethyl-p-phenylenediamine (DPD) method on a Hach DR2700 spectrophotometer. A stock solution of PAA was prepared at a concentration of 1000 mg/L as PAA, which also contained approximately 120 mg/L H_2_O_2_. To simulate a realistic scenario, H_2_O_2_ was not quenched from the stock solution since commercial PAA-based products are mixtures of PAA in equilibrium with H_2_O_2_. The dosages of PAA throughout this study are reported in terms of mg/L as PAA, but it must be understood that there is an additional 12–15% (by mass) of H_2_O_2_. During trials involving PAA, 2 mL aliquots of sample were diluted in 8 mL Milli-Q water for PAA measurement. To quench residual PAA, all collected samples were immediately transferred to vials containing sodium thiosulfate with a final concentration of 30 mg/L. PAA and H_2_O_2_ were purchased from MilliporeSigma (Oakville, ON, Canada).

Microcystin-LR and -RR, purchased from Cedarlane Labs (Burlington, ON, Canada), were measured using liquid chromatography-triple quadrupole mass spectrometry (LC-MS/MS) (Agilent 6460 Triple Quadrupole LC/MS, CA, USA) equipped with an Agilent Poroshell 120 EC-C18 column. Internal standards were added to all samples at a concentration of 5 µg/L. The analysis was carried out following the method described by Almuhtaram and Hofmann [14]. Internal standards for MC-LR (15N10, 97%) and MC-RR (15N13, 98%) were purchased from Cambridge Isotope Laboratory, Inc (Andover, MA, USA).

Dead, damaged, and live cells were identified using simultaneous staining with fluorescein diacetate (FDA) and propodium iodide (PI) stains for live and dead cells, respectively [31,32]. PI and FDA stains were purchased from ThermoFischer Scientific (Mississauga, ON, Canada).A dead-cell solution was prepared by subjecting a cell suspension to a commercial bleach containing 6% sodium hypochlorite for 60 min. Calibration standards were prepared by mixing dead and live cell suspensions to achieve dead cell percentages of 0, 25, 50, 75, and 100%. A 3 μM PI solution was prepared by diluting the PI stock solution by a factor of 500 in a staining buffer (Tris buffer at pH 7.4, 150 mM NaCl, 1 mM CaCl_2_, 0.5 mM MgCl_2_, 0.1% Nonidet P-40). FDA was prepared by dissolving 5 mg in 1 mL of acetone then diluted by a factor of 100 in phosphate-buffered saline. Then, 100 μL samples of each stain were added to each cell sample. Samples were incubated for 15 min at room temperature prior to analysis. Samples were analyzed using flow cytometry (BD-LSRFortessa™ X-20 II, BD Biosciences, CA, USA) with 525/50 and 710/50 bandpass filters for FDA and PI, respectively. The total number of events recorded in all regions was 10,000. Flow cytometry raw data were analyzed using Flow Jo and plotted using pseudocolor plots, where individual cells were categorized into 3 main categories based on the detected stains: dead, damaged, and live. Figure 7 shows 1 × 10^6^ *M. aeruginosa* cells/mL under different staining conditions. Areas with lower cell densities are represented by blue and green, while red and orange indicate regions with high cell densities. Dead cells were stained with PI because they were either metabolically inactive (no enzymatic activity) or had no DNA binding sites available. Dead cells appear in the bottom right area represented by Q3. Live cells with intact membranes were stained with FDA and appeared on the upper left area represented by Q1. Cells with damaged membranes were stained with both PI and FDA, and they appeared in the upper right area represented by Q2. The loss of membrane integrity in dead and damaged cells can potentially cause intracellular toxins to be released. Debris that is stained neither by PI nor by FDA appeared on the lower left area represented by Q4, which is excluded from the results. The number under Q represents the percentage of cells/events falling in each quadrant. Upon cell harvesting and preparation, 30–45% of the cells were naturally dead in the solution.

### 4.3. PAA Oxidation of Dissolved MC-LR and MC-RR

Solutions containing MC-LR or -RR at 50 μg/L were prepared in 100 mL of Milli-Q water. This concentration was selected to be representative of that which may occur during a bloom that is dominated by species with high microcystin content [33]. PAA was dosed at 10 mg/L, and the pH was adjusted to 6.0, 7.0, or 8.0 using 10 mM potassium phosphate buffer. Control trials were included that did not contain PAA. All trials were conducted in duplicate.

### 4.4. PAA Oxidation of M. aeruginosa Cell Culture

PAA stock solution was used to dose 10 mg/L PAA in a 100 mL suspension containing 1 × 10^6^ *M. aeruginosa* cells/mL. The addition of PAA reduced the pH of the solution to approximately 4.4, so it was adjusted using phosphate buffer to pHs 5.2, 7.0, and 8.5. At pH 5.2, where cells were observed to lyse more rapidly compared to the neutral and alkaline conditions, PAA was additionally dosed at 2 and 5 mg/L to assess the impact of a range of dosages. The solutions were continuously stirred using magnetic stir bars. Samples for cell viability, residual PAA, extracellular MC-LR, and total MC-LR measurements were collected throughout the trial durations. Total contact times with PAA were determined based on the anticipated rate of reaction. The total reaction time was 60 min at pH 4.4, 180 min at pH 5.2, and 24 h at pH 7.0 and 8.5 because of the observed slower reactions at pH levels close to the PAA pKa (8.2). For determining the concentrations of intracellular and extracellular MC-LR, 1 mL samples were filtered through 0.22 μm sterile polyvinylidene fluoride (PVDF) Millex GV syringe filters to remove all cells (therefore leaving only extracellular MC-LR) and stored at −18 °C. Parallel 1 mL samples were subjected to three freeze–thaw cycles to lyse the cells prior to filtration and stored at −18 °C for total MC-LR measurement. The difference between the total and extracellular MC-LR measurements from the parallel samples is intracellular MC-LR. For flow cytometry, 1 mL samples were collected in falcon tubes and centrifuged, and the supernatant was discarded prior to the addition of staining buffers. Control trials in the absence of PAA were conducted to compare natural versus oxidation-induced cell lysis. All trials were conducted in triplicate; however, total and extracellular MC-LR were only quantified for two trials.

## Figures and Tables

**Figure 1 toxins-16-00328-f001:**
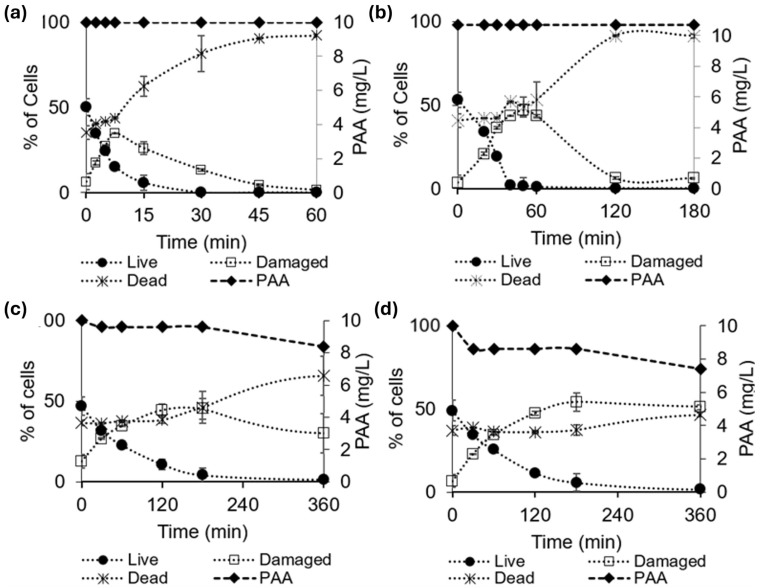
The percentages of live, damaged, and dead cells in 1 × 10^6^ cells/mL *M. aeruginosa* suspension exposed to 10 mg/L PAA at pHs (**a**) 4.4, (**b**) 5.2, (**c**) 7, and (**d**) 8.5 using 10 mM phosphate buffer. Error bars represent the standard deviations of triplicate trials.

**Figure 2 toxins-16-00328-f002:**
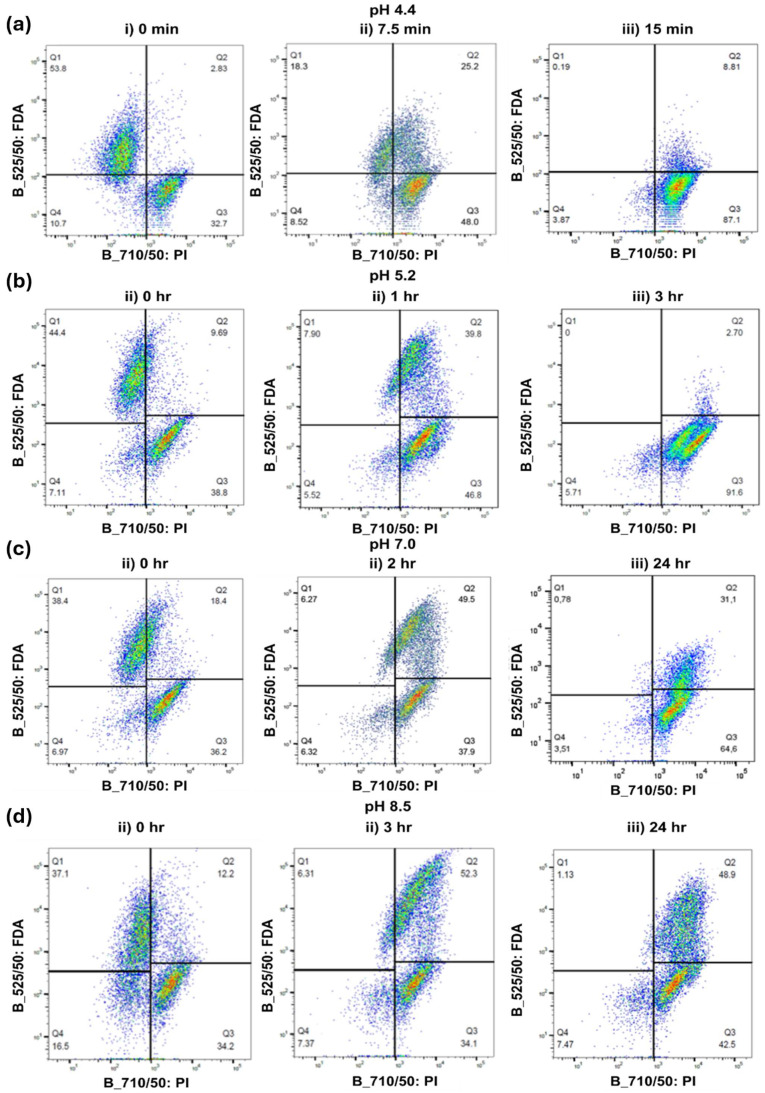
Flow cytometry pseudocolor plots of *M. aeruginosa* suspension exposed to 10 mg/L PAA at pHs (**a**) 4.4, (**b**) 5.2, (**c**) 7, and (**d**) 8.5. Analysis was performed using Flow Jo v10.8.

**Figure 3 toxins-16-00328-f003:**
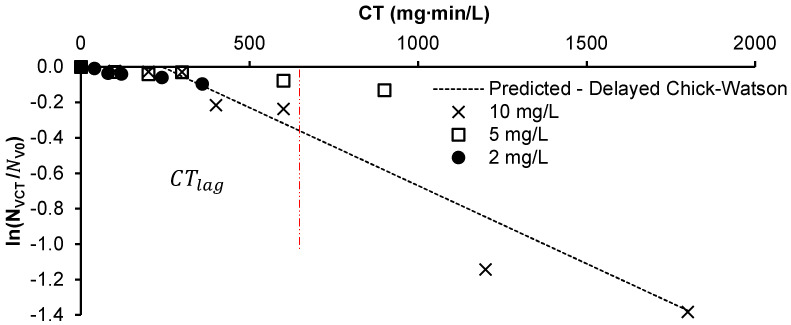
The inactivation of 1 × 10^6^ cells/mL *M. aeruginosa* exposed to 2, 5, and 10 mg/L PAA at pH 5.2 fitted using the delayed Chick–Watson model.

**Figure 4 toxins-16-00328-f004:**
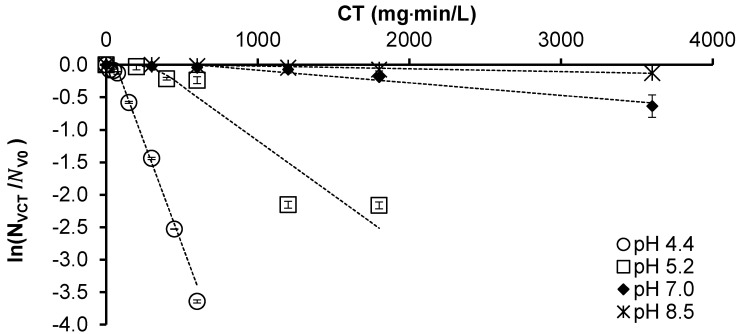
The inactivation of 1 × 10^6^ cells/mL *M. aeruginosa* at pHs 4.4, 5.2, 7.0, and 8.5. Markers represent the observed data, while the dashed lines correspond to the fitted delayed Chick–Watson model. Error bars represent standard deviations of triplicate trials.

**Figure 5 toxins-16-00328-f005:**
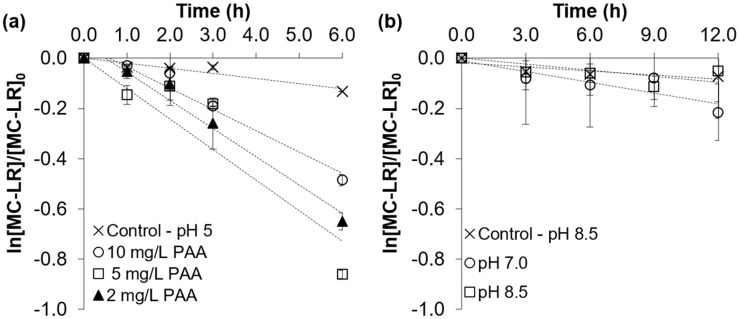
The degradation of total MC-LR in 1 × 10^6^ cells/mL *M. aeruginosa* suspension exposed to PAA with doses of (**a**) 2, 5, and 10 mg/L and the control at pH 5.2 and (**b**) 10 mg/L at pH 7.0 and pH 8.5 and the control at pH 8.5. Error bars represent duplicate trials.

**Figure 6 toxins-16-00328-f006:**
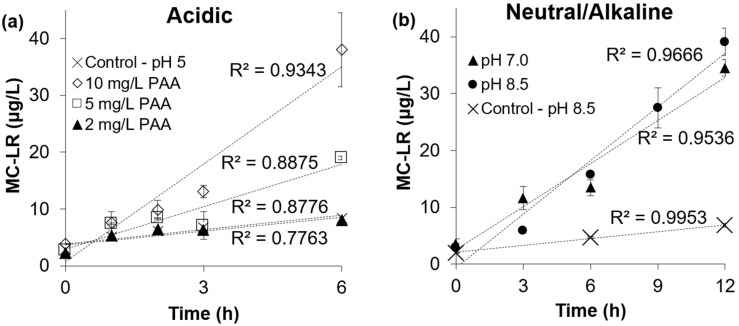
Intracellular MC-LR released from 1 × 10^6^ cells/mL *M. aeruginosa* cell suspension exposed to PAA with doses of (**a**) 2, 5, and 10 mg/L and the control at pH 5.2 and (**b**) 10 mg/L at pH 7.0 and pH 8.5 and the control at pH 8.5. Error bars represent duplicate trials.

**Figure 7 toxins-16-00328-f007:**
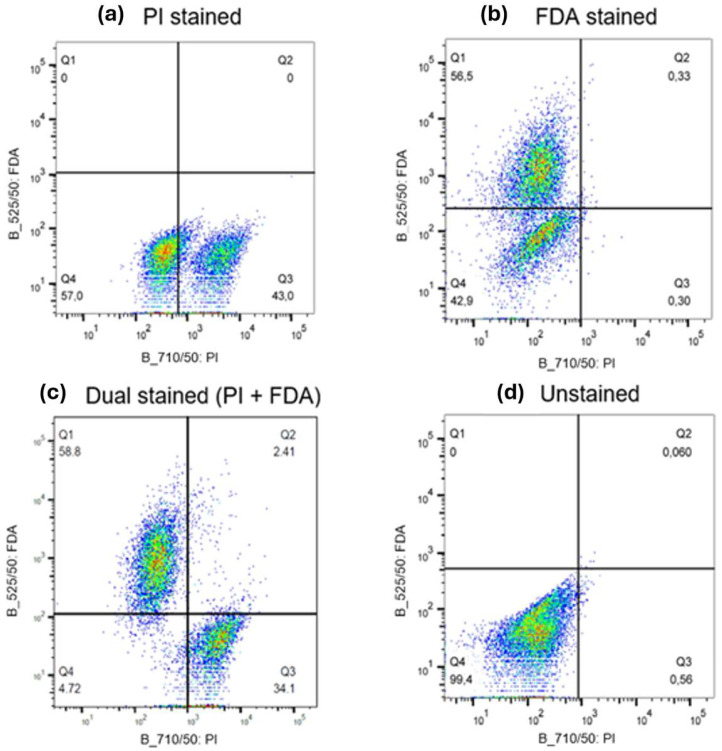
Flow cytometry plots of *M. aeruginosa* suspension stained with (**a**) PI, (**b**) FDA, (**c**) or PI+FDA or (**d**) unstained. Blue and green colors represent areas with lower cell densities, while red and orange indicate regions with high cell densities.

**Table 1 toxins-16-00328-t001:** Second-order reaction rate coefficients of MC-LR and -RR with PAA (including H_2_O_2_) and free chlorine (Cl_2_) at pHs 6, 7, and 8.

pH	k (M^−1^s^−1^)			
	MC-LR		MC-RR	
	PAA	Cl_2_ [15]	PAA	Cl_2_ [15]
6	0.004	127.8	0.002	130.3
7	0.006	91.5	0.002	90.6
8	0.265	33.1	0.2	33.8

**Table 2 toxins-16-00328-t002:** Reaction rate coefficients based on cell death for 10 mg/L PAA and 1.0 × $10^{6}$
*M. aeruginosa* cells/mL at pHs 4.4, 5.2, 7.0, and 8.5.

pH	$k_{1}$ $({\left(\mathrm{mg}/L\right)}^{-1}\cdot \min^{-1})$	$k_{1}$ $(M^{-1}\cdot \min^{-1})$	$CT_{lag}$ (M·min)	$CT_{lag}$ (mg·min/L)	R^2^
4.4	$6.3\times 10^{-3}$	480	$9.8\times 10^{-4}$	75	0.99
5.2	$1.6\times 10^{-3}$	128	$3.9\times 10^{-3}$	297	0.89
7.0	$2.0\times 10^{-4}$	15	$1.6\times 10^{-2}$	1217	0.97
8.5	$4.3\times 10^{-5}$	3.3	$2.4\times 10^{-2}$	1825	0.93

## Data Availability

The raw data supporting the conclusions of this article will be made available by the authors on request.

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
