# Peer review of "Oxidation of Microcystis aeruginosa and Microcystins with Peracetic Acid"

_toxins, 2024, doi:10.3390/toxins16080328_

Round 1

Reviewer 1 Report

Comments and Suggestions for Authors

I wonder why the authors chose to work on this subject when literature exists on the poor efficiency of the compound in controlling cyanobacterial blooms. Could you add a strong justification? Having presented more data to reinforce previous studies I suggest they include an analysis on the cost-benefit analyses of this product on a large scale with information on how long can the effects be sustained? The second doubt is about the lack of information on the effect of the chemical on higher trophic levels.

Comments on the Quality of English Language

Acceptable

Reviewer 2 Report

Comments and Suggestions for Authors

General: interesting experimental work with global producer of toxins.

Some specific questions:

Did the culture continue to produce toxins

First time a Latin Binomial is used in a scientific writing, the authority for that taxon needs to be provided; every time the name is used, it must be italicized see lines 326 and 338?

Figure 1, it seems each test started with 50 % live cells and 40 % dead, please clarify as it seems the text is written starting with 100% live and following what happened through time

In figures captions: Fig 6 /Error bars 265 represent duplicate trials.’ How did you average 2 values? in materials and methods it is stated that everything was done in triplicate?

Abstract

Clarify…Not sure what is meant by ’Microcystis aeruginosa inactivation’? Did you kill the cells? Pushed them to heterotrophy? Did not perform photosynthesis? or what

This is a bacterium, maybe they can survive without photosynthesis!

Introduction

How do you ‘degrade cyanobacteria’?

Lines 41-43: reword? Not sure what are you saying in the second part of the sentence?

’Hydrogen peroxide (H2O2) alone is considered a weak oxidant, but for contact times spanning days it has been demonstrated to lyse cyanobacteria cells’

Do you mean cells soaked in pure H2O2 for days lysed? Was the peroxide diluted?

Materials and Methods

Line 291: At 10x it is not possible to count M. aeruginosa cells, do you mean clusters/colonies?  How will this change your results?

Figures

Fig 5: in caption there is stated d b) PAA dosage of 10 mg/L at pH 7 and 8.5.’ but the figure shows ‘Control - pH 8.5, pH 7.0, pH 8.5’. Which one is correct?

Fig 6: in caption there is stated ’b) with doses 0 and 10 at pH 7.0 and 8.5.’ but the figure shows ‘pH 7.0, pH 8.5, Control - pH 8.5’. Which one is correct?

Fig 7 needs to be update in caption with the colors designation. What are the numbers under Q1- Q4? Do you mean you excluded all cells from Q4? For 7d that is 99.4 % of your data (if that is what the numbers show? I am guessing?)? What do the x and y stand for: ‘B_525/50 FDA’ and ‘B_710/50 PL’; what are the plotted solid lines showing? Finally y axis numbers for 7c seem smaller and hard to read?

In Supplementary Materials:

You have 2 figures S1?

Figure S1 Degradation of 50 μg/L of MC-LR and -RR in Milli-Q water using 10 mg/L of PAA. Error  bars represent duplicate trials.

Figure S1 Degradation of 50 μg/L of MC-LR and -RR in Milli-Q water using 1.5 mg/L of H2O2. Error bars represent duplicate trials

Check formatting in text and in references as References have a number x and the same number in [x ]

Comments on the Quality of English Language

English is OK
